# Majority sensing in synthetic microbial consortia

Razan N. Alnahhas [1], Mehdi Sadeghpour[1,2], Ye Chen [3], Alexis A. Frey[1], William Ott [2], Krešimir Josić[2,4] & Matthew R. Bennett [1,5✉]

As synthetic biocircuits become more complex, distributing computations within multi-strain microbial consortia becomes increasingly beneficial. However, designing distributed circuits that respond predictably to variation in consortium composition remains a challenge. Here we develop a two-strain gene circuit that senses and responds to which strain is in the majority. This involves a co-repressive system in which each strain produces a signaling molecule that signals the other strain to down-regulate production of its own, orthogonal signaling molecule. This co-repressive consortium links gene expression to ratio of the strains rather than population size. Further, we control the cross-over point for majority via external induction. We elucidate the mechanisms driving these dynamics by developing a mathematical model that captures consortia response as strain fractions and external induction are varied. These results show that simple gene circuits can be used within multicellular synthetic systems to sense and respond to the state of the population.

[1] Department of Biosciences, Rice University, Houston, TX, USA. [2] Department of Mathematics, University of Houston, Houston, TX, USA. [3] CAS Key Laboratory of Quantitative Engineering Biology, Shenzhen Institute of Synthetic Biology, Shenzhen Institutes of Advanced Technology, Chinese Academy of Sciences, Shenzhen 518055, China. [4] Department of Biology and Biochemistry, University of Houston, Houston, TX, USA. [5] Department of Bioengineering, Rice University, Houston, TX, USA. ✉email: matthew.bennett@rice.edu

Microbial consortia are widespread in nature, from the gut microbiome to soil microbial communities. While competition among strains within a consortium occurs, cooperation and communication can also make the consortium more adaptive and allow for more complex phenotypes than possible in monoculture[1,2]. Synthetic biologists aim to harness these advantages by moving towards synthetic microbial consortia for bioprocessing[1,3–6], and microbial ecologists are using engineered microbial ecosystems as models for studying natural microbial ecosystems[7–10].

The relative prevalence of individual species within a consortium can have major effects on the behavior of the community[10,11]. Therefore, many synthetic gene circuits have been developed to control the composition of an engineered microbial consortium[12–14]. Stable microbial strain ratios in a consortium have been achieved through cooperation between strains[14,15], selective environments[9], engineering 'self-limiting' growth[16,17], and the spatial separation of strains[11,18].

Other approaches to regulating microbial consortia include coordinating gene expression. Such coordination via intercellular communication[19–21] is most commonly achieved using quorum sensing (QS) molecules[22,23]. Existing multicellular gene circuits in microbial consortia allow for oscillations[24], DNA cycling[25], memory[26], and pattern formation[27]. In addition, consortia have been engineered to incorporate cross-feeding across strains[28] and to display social interactions[29].

The traditional function of QS in bacteria is to activate gene expression based on overall population size[22]. Here, we develop instead a consortium in which QS molecules are used to adjust gene expression at the single-cell level in response to the ratio of strains within the consortium, rather than the overall size of the population. We engineer this consortium by designing a multicellular circuit that extends the traditional co-repressive topology[30] to two strains. Whereas the classical co-repressive toggle architecture utilizes two transcription factors that repress the production of one another, each strain within the co-repressive consortium produces a QS molecule that signals the other strain to shut down production of an orthogonal QS molecule. Specifically, QS molecules from one strain induce the expression of a repressor in the opposite strain, shutting off production of its QS molecule.

Our multicellular co-repressive circuit is modifiable and inducible. To modify the system, we engineer a variant of the co-repressive consortium that activates gene expression in the minority fraction strain. To demonstrate inducibility, we externally activate gene expression in one strain while simultaneously deactivating expression in the opposite strain across a range of strain ratios. Overall, these results will allow for the construction of complex synthetic consortia with autonomous gene regulation through the sensing of strain ratios within the consortium.

## Results

**Co-repressive consortium.** To construct the co-repressive consortium (Fig. 1), we transformed two different sets of plasmids into the same background strain of *Escherichia coli*. The two sets of plasmids each contained three genes encoding distinct elements: an acyl-homoserine lactone synthase (either RhlI or CinI), a transcriptional repressor (LacI-11 or RbsR-L), and a fluorescent reporter (sfCFP or sfYFP). In response to environmental 3OHC14 homoserine lactone (C14-HSL), the "cyan" strain upregulates the gene encoding the chimeric repressor RbsR-L[31,32], which in turn downregulates the genes encoding the super folder variant of cyan fluorescent protein (sfCFP) and RhlI (the synthase that produces C4 homoserine lactone, C4-HSL)[33]. In contrast, the "yellow" strain responds to environmental C4-HSL

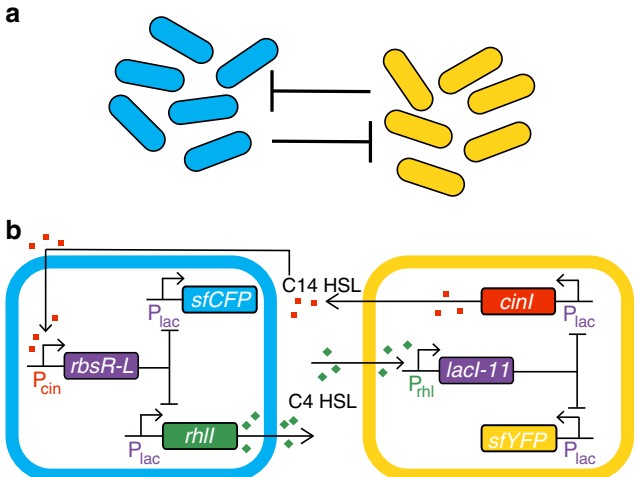

**Fig. 1 Co-repressive consortium circuit. a** The regulatory structure of the co-repressive consortium in which each strain downregulates production of QS molecule and fluorescent protein in the other strain. **b** A detailed genetic circuit diagram of the co-repressive consortium. The cyan strain produces C4-HSL via the RhlI QS synthase. C4-HSL induces the expression of LacI-11 in the yellow strain. LacI-11 shuts off CinI and sfYFP in the yellow strain. 3OHC14-HSL produced by the CinI QS synthase in the yellow strain induces expression of RbsR-L in the cyan strain. RbsR-L shuts off RhlI and sfCFP in the cyan strain.

by upregulating the gene encoding a dimeric version of LacI (LacI-11)[31], which downregulates the genes encoding the super folder variant of yellow fluorescent protein (sfYFP) and CinI (the synthase that produces C14-HSL)[34]. All of the above proteins contain C-terminal *ssrA* degradation tags that target the ClpXP protease[35]. Note that the background strain contains the genes *rhlR* and *cinR* in its genome. The proteins RhlR and CinR are the transcription activators that use C4-HSL and C14-HSL as ligands, respectively[33,34]. The plasmids containing the transcriptional repressor genes also code for AiiA, a lactonase that degrades C4-HSL and C14-HSL. With this genetic layout, each strain expresses a QS molecule when cultured alone. Therefore when cultured together, the more numerous, or majority, strain produces more QS molecule overall, eventually shutting off QS production in the less numerous, or minority, strain.

To ensure that the two strains function properly, we initially tested them in monoculture. When each strain was grown separately, in the absence of the opposite strain's QS molecule, each strain was 'ON' (expressing both QS molecule and fluorescent protein). Addition of the opposite QS molecule turned each strain 'OFF' in monoculture (Supplementary Fig. 2). Whether the strains were ON or OFF did not affect their growth rates, and the growth rates of the two strains did not differ appreciably (Supplementary Fig. 3). We thus verified the proper function of the strains and confirmed that the state of the strains (ON or OFF) does not impact growth rates.

**Majority wins pattern.** We hypothesized that when the strains are co-cultured, the majority fraction strain will produce more QS molecule and therefore repress QS production in the minority fraction strain. To test this hypothesis qualitatively, we mixed monocultures of the two strains in 10% increments from 100% cyan strain to 100% yellow strain. We then spotted each mixture onto an LB agar plate, incubated the plates overnight, and imaged the resulting colonies (see "Methods"). As shown in Fig. 2a, the majority strain, in general, repressed expression in its counterpart: Spots seeded with 70–100% cyan strain were bright cyan and

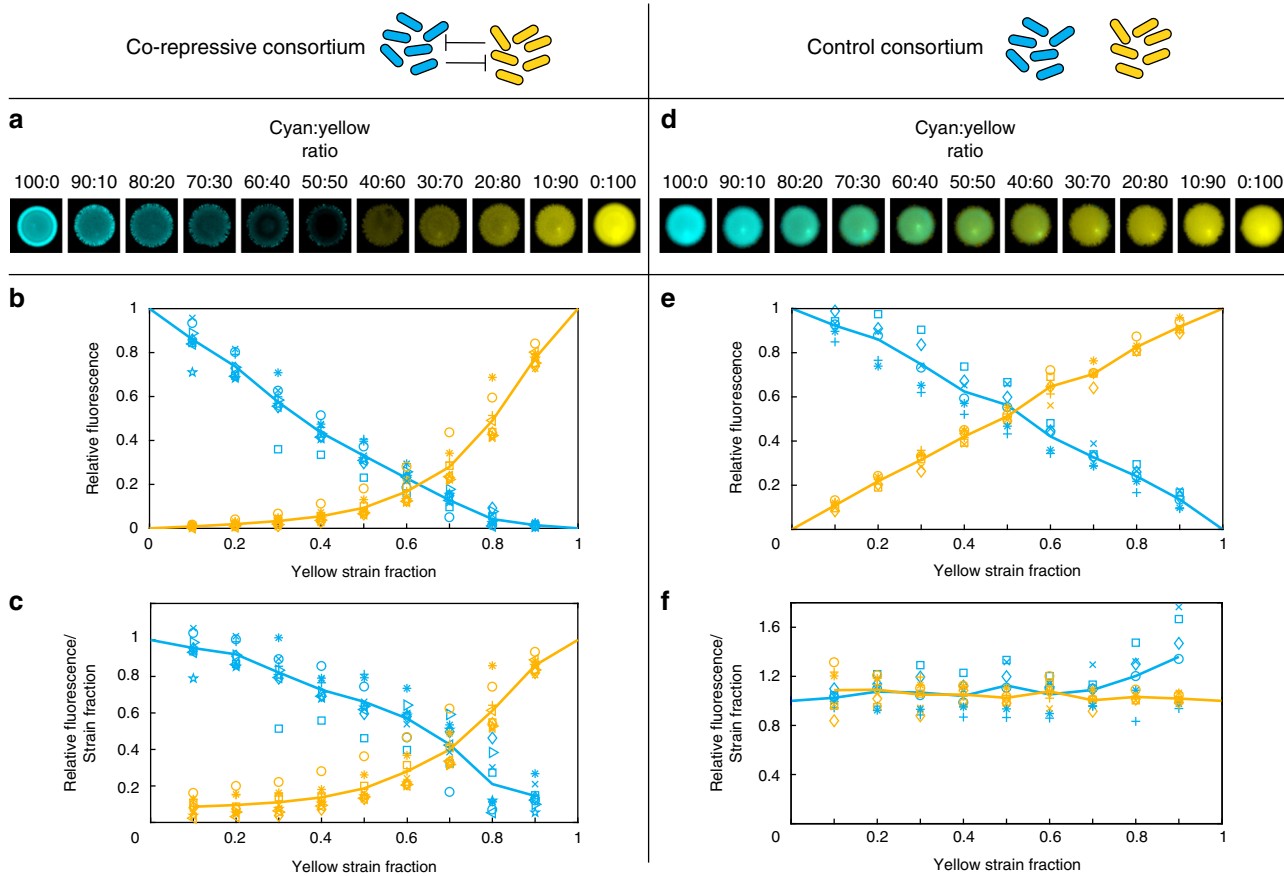

**Fig. 2 Majority wins pattern. a** Fluorescence images of spots of the co-repressive consortium on agar plates at mixtures of the two strains with 10% increments from 100% cyan strain to 100% yellow strain. **b** Fluorescence intensity of the co-repressive consortium at mixtures in liquid cultures in a 96-well plate ($n = 9$). Results were obtained using 10% increments starting with 100% cyan strain and ending with 100% yellow strain. To compare fluorescence intensities with a maximum, they are normalized to the values measured in wells containing a single strain of the respective color (100% wells). Fluorescence intensity correlates with strain fraction in a majority wins pattern. Same symbols with different colors represents the yellow and cyan intensities from the same well of the same replicate experiment and lines represent averages across all replicates (consistent throughout all graphs presented). **c** The data from (**b**) normalized to strain fraction to account for any change in fluorescence due to the decrease in the number of cells containing the fluorescence gene. The co-repressive consortium still follows a majority wins pattern even after normalization to strain fraction suggesting active shut off of the minority strain. **d–f** Same as (**a–c**) but for a control consortium lacking intercellular signaling and with constitutive expression of *sfcfp* or *sfyfp* in the respective strains ($n = 6$). The control consortium does not follow a majority wins pattern after normalization to strain fraction. Source data are provided as a Source Data file.

had little to no yellow fluorescence, suggesting active shut off of the yellow strain. Spots seeded with 70–100% yellow strain were bright yellow and displayed little to no cyan fluorescence, suggesting active shut off of the cyan strain. In the middle, with 40–60% yellow strain, the spots displayed dim cyan to dim yellow.

To quantify this pattern, we grew the same 10% increment ratios but in liquid culture in 96-well plates. This allowed for higher throughput using a plate reader, quicker measurement of replicates, and more accurate comparison of replicates by examining data once the cells reach the same OD. Results from this experiment are shown in Fig. 2b. The yellow and cyan fluorescence intensities for wells with mixed strains are normalized to each fluorescence maximum (the well with 100% yellow or cyan strain). The co-repressive strains presented a majority wins fluorescence pattern, as shown in Fig. 2b. This pattern is consistent with that of the spots in that there is bright cyan fluorescence in wells with over 70% cyan strain fraction, bright yellow fluorescence in wells with over 70% yellow strain fraction, and dim cyan and yellow fluorescence in wells with 40–60% mixtures of the two strains (both fluorescence levels below the half-maximum). To account for any change in fluorescence due

to a difference in the number of cells possessing the *sfcfp* or *sfyfp* gene in each well, we normalized each fluorescence to the strain fraction of the same color. As seen in Fig. 2c, the majority wins fluorescence pattern is maintained in the co-repressive consortium even after normalization to strain fraction. This suggests that the change in fluorescence is not simply due to the change in number of cells expressing either fluorescent protein and suggests active downregulation of gene expression in the minority strain and upregulation of gene expression in the majority strain.

To validate that this system is using QS to respond to strain ratios and not population size, we performed the same experiments in deep well plates with larger culture volume and obtained the same results (Supplementary Fig. 4, source data are provided as a Source Data file.). To confirm that the final strain ratios were the same as the initial, mixed strain ratios, we performed serial dilutions of the cultures at the end of the experiments and plated them onto LB agar plates. We then counted the number of resultant yellow and cyan colonies to measure the strain ratios. We did this for 3–4 of the biological replicates from each graph presented throughout the paper. Supplementary Fig. 5 shows the results of this quantification and

that the final, measured strain ratios were very close to the initial, mixed strain ratios.

We next wanted to confirm that the decrease in the fraction of cells with the *sfcfp* or *sfyfp* gene was not the main cause of the fluorescence pattern observed. We therefore performed the same set of experiments with a control consortium. The strains within the control consortium each coded for constitutive *sfcfp* or *sfyfp* expression, but the two strains did not communicate with each other. Moreover, except for the point mutations that differ between *sfcfp* and *sfyfp*, these strains were isogenic. Figure 2d shows that from 100% cyan strain to 100% yellow strain spots of the control consortium, fluorescence gradually changed from cyan to yellow without much change in overall brightness, unlike the co-repressive consortium.

Interestingly, the difference in fluorescent protein expression between the co-repressive and control consortia is not dramatic at the population level in the 96-well plate assay (Fig. 2b and e). A decrease or increase in fluorescence of the bulk culture is certain to happen simply because the relative abundance of one or the other strain is changing. However, expression in the co-repressive and control consortia differ dramatically at the single-cell level (as measured by fluorescence divided by respective strain fraction), as shown in Fig. 2c and f. Specifically, cells within the co-repressive consortium clearly respond to the relative abundances of the two strains, whereas in the absence of cell–cell signaling, the average single-cell fluorescence is independent of strain fraction.

We next asked whether we could tune the responsiveness of the majority wins system by introducing external inducers into the media. To do so, we first grew the same 10% increments of the co-repressive strains with 10 mM ribose or 10 mM IPTG to fully induce the cyan or yellow strain, respectively. Addition of ribose induces *rhlI* and *sfcfp* in the cyan strain by relieving repression via RbsR-L. On the other hand, addition of IPTG induces *cinI* and *sfyfp* in the yellow strain by relieving repression via LacI-11. Figure 3a shows that 10 mM of either inducer turned ON the appropriate strain (and turned OFF its counterpart) across all ratios tested (except when one strain was absent).

Next, we tested if we could more finely tune the cross-over point of the relative fluorescences in the system by introducing various amounts of inducer. To do this, we grew the same 10% increments of the co-repressive consortium with decreasing amounts of IPTG. Figure 3b shows the response of the system to decreasing concentrations of IPTG across the strain ratios. A concentration of 0.5 mM IPTG was sufficient to turn ON the yellow strain (and turn OFF the cyan strain) at all yellow strain fractions above 10%. At 0.1 mM IPTG, the yellow strain is ON at all fractions higher than about 40%, and at 0.05 mM IPTG this threshold increased to about 60% yellow strain. With 0.01 mM IPTG, responses were close to those observed with no inducer (Fig. 2c), as low IPTG concentrations have little to no effect on the system. This pattern shows that increasing amounts of inducer shift the cross-over point, and that the more outnumbered the strain is, the more inducer is needed to counteract the repression from the more numerous strain.

**Mathematical modeling of co-repressive mechanisms**. To understand the mechanisms that drive the observed behaviors of the consortium, we created a mechanistic mathematical model of the system. A summary of the model is given in the "Methods" section, while a full description is provided in the Supplementary Methods. We extended existing models of single-cell biological switches[30] in order to capture the dynamics of interacting bacterial communities. Since we are looking at the population-level fluorescence state of the system, we neglected single-cell level dynamics and cell to cell variability. The model builds on one we developed previously to describe a co-repressive consortium[36]. Here we added the effects of inducer and leakiness, while neglecting the effects of metabolic loading. Due to the symmetry of the system, we only modeled the induction of the yellow strain. A similar model for the effect of the inducer on the inhibition of the protein synthesis by a QS signal was previously studied by Salzano et al.[37].

Figure 4 compares model solutions to experimentally measured relative fluorescence intensities as a function of IPTG concentration for several different yellow strain fractions. Similar to the experimental data, the model results show the fluorescence at a time when the total cell count reached 85% of the carrying capacity of the well. The model qualitatively captures the experimental behavior of the co-repressive consortium. In particular, at a fixed ratio of the yellow strain, increasing IPTG concentration increases yellow fluorescence. Therefore, even for relatively small fractions of the yellow strain, the yellow strain fluorescence 'wins' over the cyan strain fluorescence, provided enough IPTG is present.

We made several simplifying assumptions when constructing our model. We did not include the synthesis, degradation, and diffusion of the signaling molecules. Instead, we assumed that the QS signal is proportional to the total number of synthase proteins in each strain. As noted before, we did not model cell to cell variability because of our focus on strain-wide average fluorescence states. The model did not explicitly include the number of cells as a parameter, predicting that behavior should be independent of colony size. This prediction was confirmed in separate experiments (Supplementary Fig. 4).

Thus despite simplifications, our model captures the behavior of the system well. This suggests that the main mechanisms that determine the average dynamical behavior of the consortium are co-repression by QS signals, cell growth, enzymatic degradation of the proteins, and inducer concentration.

**Minority wins consortium**. The majority wins pattern consortium provides a multicellular system in which genes turn on in the more numerous strain. We next engineered a consortium in which strains turn on gene expression when they are in the minority. To engineer such a 'minority wins' consortium, we changed the promoters of *sfcfp* and *sfyfp* in the co-repressive consortium described above (Fig. 5a). In the minority wins consortium, the genes encoding sfCFP and sfYFP are induced directly by the QS molecule produced by the opposite strain. The opposite strain's QS molecule still induces expression of the *rbsR-L* and *lacI-11* genes in the respective strains in order to repress the *rhlI* and *cinI* genes, respectively. With these modifications, the strains repress each other's ability to produce QS molecule, but require the other strain's QS molecule to express sfCFP or sfYFP. In order to be fluorescent, each strain must therefore be in the minority so that the opposite strain is producing QS molecules.

In this system, when either strain is cultured alone it should express QS molecules, but it should not express fluorescent protein. To verify this prediction, we tested each strain independently and confirmed that they did not fluoresce in monoculture (Supplementary Fig. 7). We also confirmed that the opposite strain's QS molecule induced fluorescence in monoculture (Supplementary Fig. 7).

To test the minority wins consortium we performed the same 96-well plate experiments done on the majority wins consortium previously. We mixed monocultures of the two strains in 10% increments from 100% cyan strain to 100% yellow strain, without IPTG or ribose, and measured fluorescence in a plate reader. The results shown in Fig. 5b display the resulting minority wins pattern of this consortium. Since the strains do not fluoresce

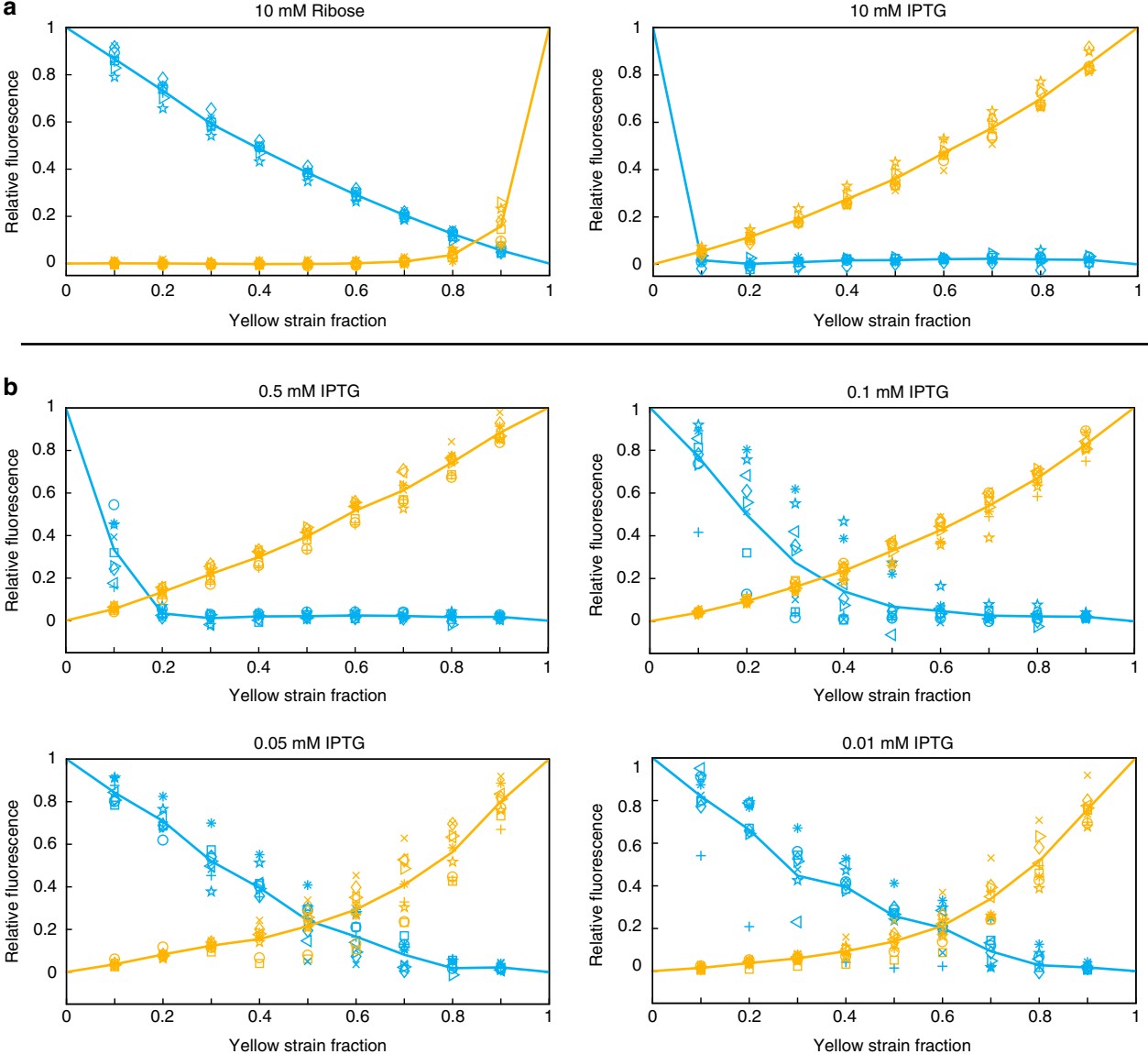

**Fig. 3 Inducer effect on the majority wins pattern. a** Relative cyan (blue curve) and yellow (yellow curve) fluorescence as a function of strain fraction in the presence of 10 mM ribose (left) or 10 mM IPTG (right) ($n = 9$). Full induction with 10 mM ribose or IPTG induces the respective strain regardless of strain fraction. Fluorescences are normalized to maximum values from wells containing a single strain of the respective color (100% wells). Normalization of these data to strain ratio is presented in Supplementary Fig. 6a to account for the change in number of cells containing the fluorescence gene. **b** Decreasing amounts of IPTG shift the fluorescence curves differently and alter the majority wins pattern less ($n = 9$). See Supplementary Fig. 1 for corresponding modeling data showing the modulation of cross over point with inducer. For each of the above graphs, similar symbols with different colors represent the yellow and cyan intensities from the same well of the same replicate experiment and lines represent averages across all replicates. Further, fluorescences are normalized to maximum values from wells containing a single strain of the respective color (100% wells). Normalization of these data to strain ratio is presented in Supplementary Fig. 6b to account for the change in number of cells containing the fluorescence gene. Source data are provided as a Source Data file.

when cultured alone, we could not normalize fluorescence to a maximum from the 100% well as we did with the majority wins consortium. However, we did still normalize the the fluorescence intensities to strain fraction in order to account for any change in fluorescence due to the decrease in the number of cells containing the fluorescence gene. As shown in Fig. 5b, the fluorescence of each strain is inversely related to its fraction in the consortium, as predicted. A mathematical model similar to the one describing the majority wins consortium again shows excellent agreement with the experimental data (Fig. 5c), further validating the conclusions (see Supplementary Methods for a full description of the model). Again to confirm that this system is using QS to

respond to strain ratios rather than overall population size, we performed the same experiments in deep well plates with larger culture volume and obtained the same results (Supplementary Fig. 4).

**Response to fluctuating strain fractions.** To demonstrate an application of the majority and minority wins patterns, we grew each system in an environment that allows for strain ratio fluctuations over time. To do so, we chose the 'Hallway' microfluidic device that has small cell trapping chambers off to the side of the media flow channel, as shown in Supplementary Fig. 8. This

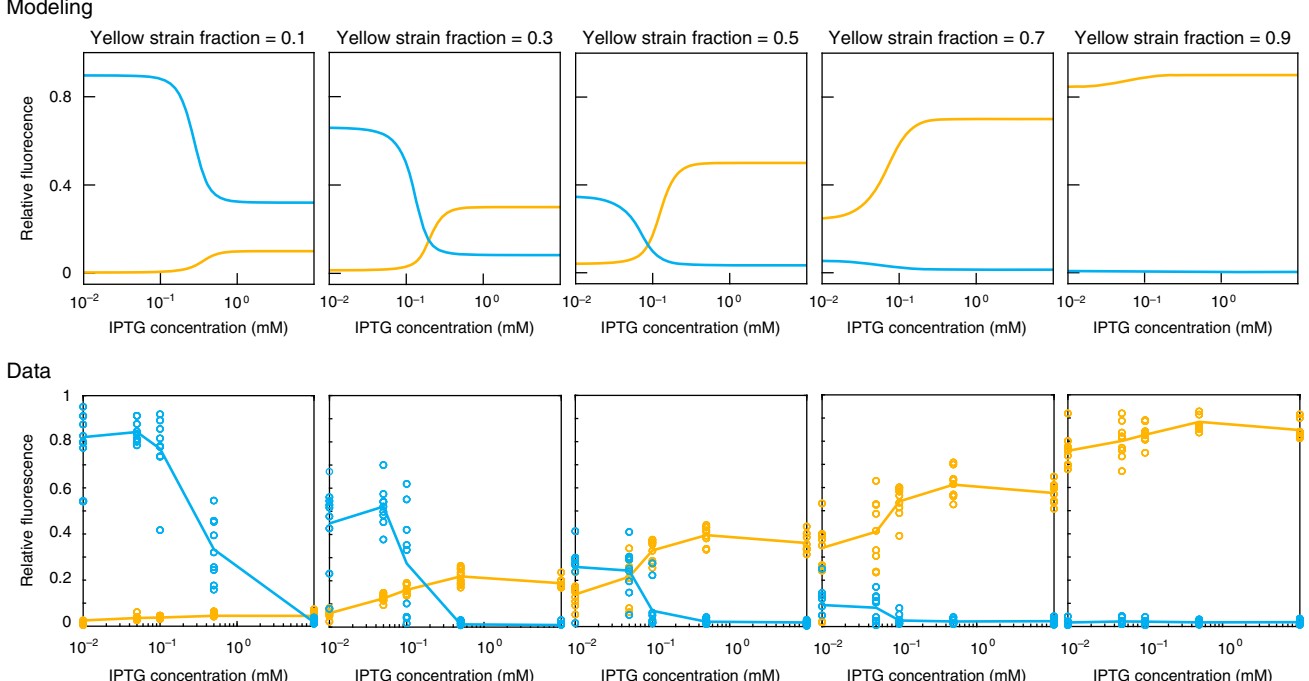

**Fig. 4 Modeling strain ratio and inducer effect on fluorescence.** The top row of graphs present relative fluorescence of the yellow and cyan strains as a function of normalized inducer concentration at set strain ratios from the mathematical model. The bottom row of graphs present relative fluorescence of the yellow and cyan strains as a function of IPTG concentration at set strain ratios from the same experimental data presented in Fig. 3. To compare fluorescence intensities, all fluorescences are normalized to the values measured in wells containing a single strain (wells containing either 100% of the yellow strain or 100% of the cyan strain). Circles represent replicate experiments and lines represent averages across all replicates. The model agrees qualitatively with experimental results across all strain ratios.

device has previously been demonstrated to result in fluctuations of strain ratios over time due to the small population size within the cell trapping region[38]. Microfluidic devices provide single-cell gene expression data by using fluorescence microscopy and continuous growth due to constant fresh media flow[39]. We mixed monocultures of both strains from either the majority or the minority wins consortium equally and then seeded many cells from this mixture into the cell traps of the microfluidic device to increase the chance of capturing both strains in each trap. We then provided the seeded cells with continuous fresh media without IPTG or ribose and imaged the cells over time. Figure 6 shows one representative example from each consortium in which both strains were captured.

We observed that the ratio of the two strains fluctuated over time, and that the fluorescence state responded accordingly. In the majority wins example shown in Fig. 6a, the trap initially had more cells of the cyan strain and therefore the cyan cells were fluorescent, and the cells of the yellow strain were not. As time went on, the yellow strain began to take over more of the trap, and began turning on sfYFP and QS molecule expression as a result. This occurred because as the yellow strain began to dominate the trap, the number of cyan cells was reduced and hence the concentration of the cyan strain's QS molecule decreased. With this reduction in QS molecule, the expression level of LacI-11 in the yellow strain decreased enough to allow for production of QS molecule and sfYFP in the yellow strain. As the yellow strain continued to dominate and produce QS molecule, eventually the concentration of QS molecule increased enough to shut off production of QS molecule and sfCFP in the cyan cells.

In the minority wins example shown in Fig. 6b, initially the cyan strain was in the minority and therefore fluorescent due to production of QS molecule by the majority fraction yellow strain. As the cyan strain began to take over the trap, it's fluorescence

decreased because there were fewer yellow strain cells producing the QS molecule that the cyan strain needs to turn on sfCFP expression. Once the cyan strain took over enough of the trap to become the majority, it began producing enough QS molecule to turn on sfYFP expression and off QS molecule production in the yellow cells. This resulted in the final image where yellow was in the minority and fluorescent and cyan was in the majority and not fluorescent.

These results show that not only do both consortia display gene expression patterns that depend on the strain ratios, they also dynamically respond to changes in this ratio over time. Videos of the full data displayed in Fig. 6 are provided in the Supplementary Information (Supplementary Videos 1 and 2). Quantification of these data depend on estimating the strain ratios when the strains are not fluorescent by using dynamic thresholds for fluorescence. The graphs in Fig. 6 display the quantification of the two representative examples shown; similar graphs from other replicates are presented in Supplementary Fig. 9. For the majority wins consortium, the fluorescence patterns followed the strain ratio patterns, while for the minority wins consortium, the fluorescence patterns were inversely related to the strain ratio patterns. Videos of the full data displayed in Supplementary Fig. 9 are provided in the Supplementary Information (Supplementary Videos 3–8).

## Discussion

Here we have built two types of co-repressive consortia that alter gene expression based on the ratio of their constituent strains and respond to changes in the ratio over time. This is a useful phenotype for a wide range of engineered consortia in which strains need to adjust their gene expression in response to changes in the strain composition within a consortium. We have developed two

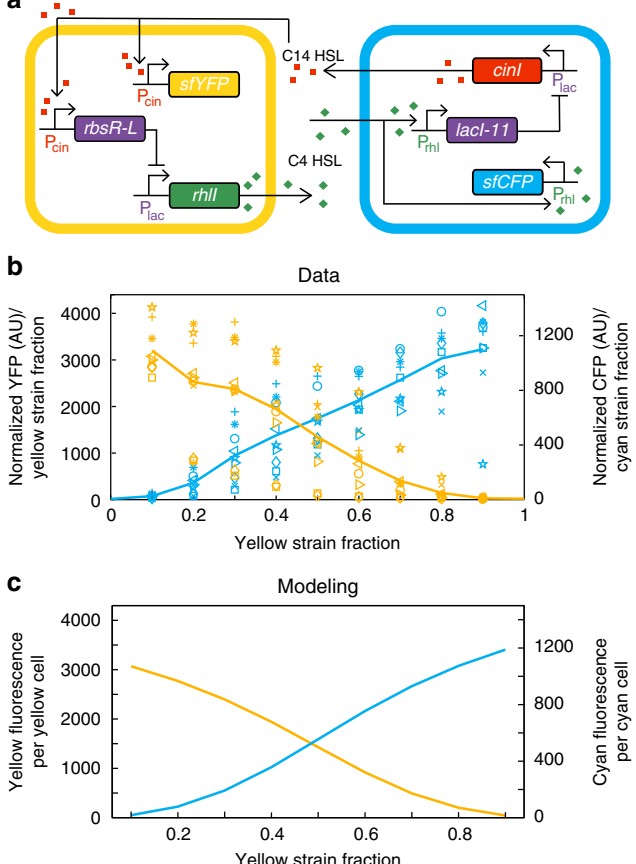

**Fig. 5 Minority wins pattern. a** Genetic circuit of the minority wins consortium. **b** Quantification of the fluorescence intensities of the minority wins consortium at mixtures of the two strains with 10% increments from 100% cyan strain to 100% yellow strain from liquid cultures in a 96-well plate ($n = 9$). Data are normalized to strain fraction to account for any change in fluorescence due to the decrease in the number of cells containing the fluorescence gene. Same symbols with different colors represent the yellow and cyan intensities from the same well of the same replicate experiment and lines represent averages across all replicates. Source data are provided as a Source Data file. **c** Modeling of minority wins consortium fluorescence as a function of yellow strain fraction. The model and data qualitatively agree, and both demonstrate the inverse relationship between strain fraction and fluorescence.

types of ratiometric sensors, one in which genes are turned on when a strain is more numerous in the consortium and one where genes are turned on when a strain is less numerous in the consortium. Possible applications of the majority wins phenotype include population control by encoding a toxin gene within the circuit. Then if a strain begins to overgrow and become a majority, the toxin could be expressed and reduce the growth rate of this strain until a balance is reached again. A possible use of the minority wins phenotype would be in bioprocessing. If a biochemical pathway is split between strains within a consortium and one strain begins to overgrow there would be a buildup and bottleneck in the pathway. If the minority wins genetic circuit is adapted to this pathway, then the system would downregulate gene expression in the overgrowing strain to prevent a buildup while also upregulating gene expression in the minority strains preventing a backup at that step of the process. We have also demonstrated that the co-repressive consortium can be induced to turn on genes in either strain as desired despite the current strain ratio. Our mathematical modeling confirms the mechanism

underlying these dynamics, and shows that they are determined by the relationship between strain ratio and inducer concentration. The model also captures gene expression states at different strain ratios and inducer concentrations. As the types and functions of engineered consortia become more complex, we need more mechanisms to control gene expression of strains within consortia. Our engineered system provides a mechanism in which gene expression state depends on stain ratios while allowing external induction of genes as needed.

## Methods

**Control consortium plasmids.** Plasmids for all strains used were constructed using traditional restriction enzyme cloning and golden gate cloning methods. Plasmids for the 'control strains' are from Alnahhas et al.[38]. The control strains are BW25113 ΔaraC ΔlacI E. coli cells[40] transformed with a single plasmid each. The plasmids are identical except for the point mutations that differ between *sfcfp* and *sfyfp*[41,42]. The fluorescence genes are under the control of a constitutive PIq promoter[43] and modified bicistronic design ribosome binding site[44] and are tagged for degradation with a mutagenized ssrA tag that ends with amino acids ASV[35]. The fluorescence genes are followed by the iGEM registry B0014 terminator, and the plasmids contain an ampicillin resistance gene and p15A origin of replication.

**Co-repressive consortium plasmids.** Majority and minority wins plasmids are listed in Supplementary Tables 3 and 4. The parts, including BCDs, promoters, degradation tags, CDS, and the use of a lactonase, were chosen from characterized use in consortia[24,45,46]. In these the degradation tags and lactonase were used to decrease timescales; we kept these in place to not alter the input–output relationships of the signaling systems which are known to work well[24]. Plasmid maps are provided in the supplement, and the plasmids (and sequences) are available on Addgene. The plasmids are transformed into the CY027[24] strain which is a BW25113 ΔaraC ΔlacI ΔsdiA E. coli strain with *cinR*[34] and *rhlR*[33] inserted at the attB site under constitutive promoters. Each strain in the majority wins consortia is transformed with three plasmids: one coding for the QS molecule enzyme, one coding for the repressor, and one coding for the fluorescence gene.

For cyan and yellow strains respectively, the QS molecule plasmids contain RhlI (ATCC #47085) or CinI (ATCC #10004) under an engineered lac promoter and bicistronic design ribosome binding site[44]. These QS systems were chosen for their lack of crosstalk in the ΔsdiA strain[24]. Both genes are tagged for degradation with a standard LAA ssrA tag[35] and are followed by the iGEM registry B0014 terminator. These plasmids contain a spectinomycin resistance gene and p15A origin of replication.

For the cyan strain, the repressor plasmid contains the RbsR-L chimeric repressor[31] under the control of an engineered cin promoter, and for the yellow strain, the repressor plasmids contains the LacI-11 dimeric repressor[31] under the control of an engineered rhl promoter. Both genes are controlled by bicistronic design ribosome binding sites[44], are tagged for degradation with a standard LAA ssrA tag[35], and are followed by by the iGEM registry B0014 terminator. Both plasmids also include the AiiA gene for QS molecule turnover[47] under the control of an engineered xylose inducible promoter and bicistronic design ribosome binding site[44] and tagged for degradation with a standard LAA ssrA tag[35]. These plasmids have a chloramphenicol resistance gene and pSC101 origin of replication.

Lastly, the fluorescence plasmids for the cyan and yellow strains contain *sfcfp* and *sfyfp*, respectively, under the control of the same engineered lac promoter and bicistronic design ribosome binding site[44] as the QS genes in the first plasmids. They are also tagged for degradation with a standard LAA ssrA tag[35] and followed by the by the iGEM registry B0014 terminator. These plasmids contain a kanamycin resistance gene and pMB1 origin with an ROP element[48].

The strains of the minority wins consortia are the same BW25113 ΔaraC ΔlacI ΔsdiA E. coli cells with *cinR*[34] and *rhlR*[33] inserted at the attB site under constitutive promoters[24] as the majority wins, and they contain the same QS and repressor plasmids described above. The fluorescence plasmids are different and contain either *sfcfp* under an engineered rhl promoter or *sfyfp* under an engineered cin promoter both with the same bicistronic design ribosome binding site[44], standard LAA ssrA tag[35], and iGEM registry B0014 terminator as described above. These plasmids also contain a kanamycin resistance gene and pMB1 origin with an ROP element[48]. The fluorescence colors are switched in the minority wins, so the cyan strain contains the CinI and LacI-11 plasmids described above, and the yellow strain contains the Rhl and RbsR-L plasmids described above.

**Spotting images.** For all experiments, we transformed the strains with plasmids and plated each onto an LB agar plate with 50 μg mL$^{-1}$ of kanamycin and spectinomycin and 33 μg mL$^{-1}$ of chloramphenicol for the co-repressive (majority or minority wins) consortia or onto an LB agar plate with 100 μg mL$^{-1}$ ampicillin for the control consortium and incubated 16–18 h at 37 C. We then grew each strain independently over night: we started with two tubes with 5 mL of LB broth containing the appropriate antibiotics and inoculated each with a colony from the transformation plate (one from the cyan strain and one from the yellow strain of

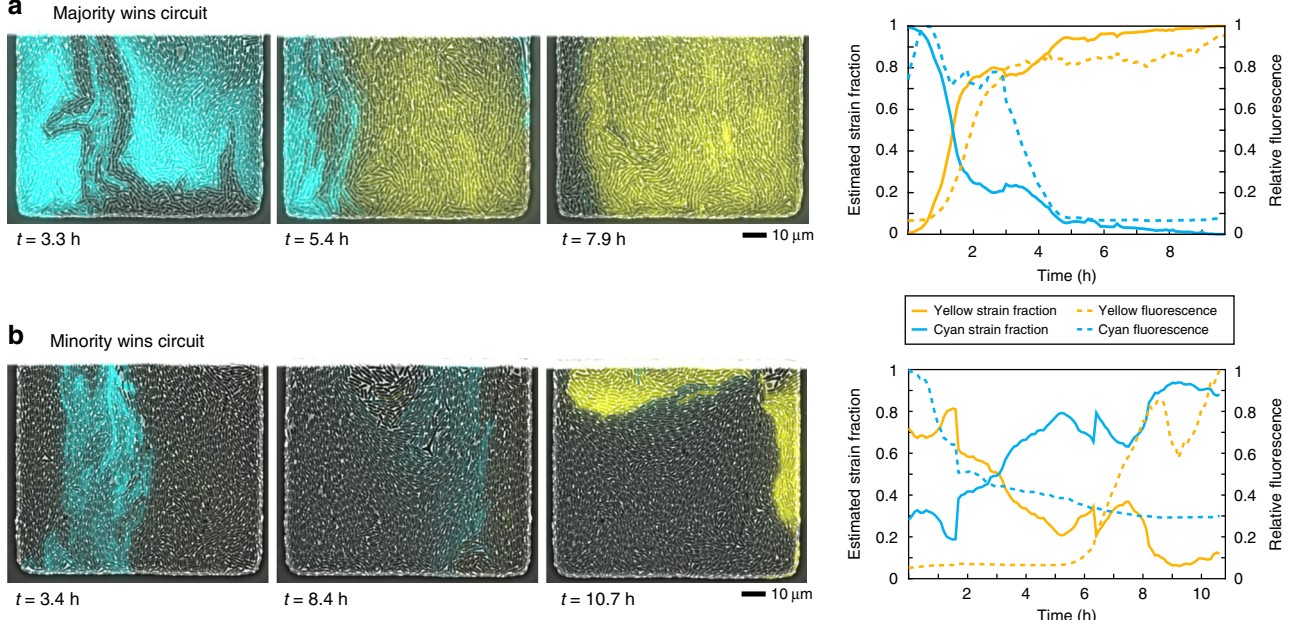

**Fig. 6 Majority and minority wins consortia in fluctuating environment. a** Response of majority wins consortium to strain ratio fluctuations in a microfluidic device over time. As the cyan strain went from the major fraction to the minor fraction of the population, the fluorescent state of the system switched from cyan strain ON to yellow strain ON, displaying a majority wins pattern as strain ratios fluctuate. To the right is a quantification of the fluorescence compared to estimated strain ratios over time. Video for this experiment is provided in the supplement as Supplementary Video 1 (Majority Wins1). **b** Response of minority wins consortium to strain ratio fluctuations in microfluidic device over time. As the cyan strain went from the minor fraction to the major fraction of the population, the fluorescent state of the system switched from cyan ON to yellow ON, displaying a minority wins pattern as strain ratios fluctuate. To the right is a quantification of the fluorescence compared to estimated strain ratios over time. Video for this experiment is provided in the supplement as Supplementary Video 2 (Minority Wins1).

the consortium being tested). These cultures were grown overnight in a shaking incubator at 37 C for 16–18 h. To generate the images in Fig. 2a and b we mixed the overnight cultures of the cyan and yellow strains in 10% increments from 100% cyan strain to 100% yellow strain with a total volume of 100 mL. Then we took 1 mL of each mixture and spotted onto a small (60 mm) petri dish with LB agar, appropriate antibiotics, and 2 mM xylose to induce AiiA expression—each spot was on its own plate to prevent interactions between spots. These plates were incubated for 12 h at 37 C and then imaged under bright field, cyan fluorescence, and yellow fluorescence channels of an upright Nikon microscope with a ×0.5 objective. Individual fluorescent images obtained from the microscope Nikon Elements program were exported into tiff files for each time, and we merged the cyan and yellow channels in ImageJ to generate each spot shown in the figure. The images of each spot were placed next to each other in Adobe Illustrator to generate the images seen in Fig. 2a and b.

**96-well plate assays**. The same method with slight variations was used to carry out all the 96-well plate assays for the data shown in Figs. 2, 3, and 5b. The transformations and overnight cultures for these experiments are performed using the same methods. In the morning, the overnight cultures we each diluted 1/200 into fresh LB with antibiotics (still monocultures). These cultures were grown to an OD of 0.5, usually about 1.5–2 h. In the mean time, the media for the 96-well plate was prepared. For the control consortium this media contained only LB and ampicillin. For the majority and minority wins versions of the co-repressive consortium, the media contained LB, antibiotics, and 2.2 mM (1.1x) xylose to induce AiiA. 180 mL of this media was placed into each well of the 96-well plate. Once the cultures reached the 0.5 OD, they were mixed in 10% increments from 100% cyan strain to 100% yellow strain with a total volume of 100 mL, and 20 mL of each mixture was used to inoculate the appropriate wells. Since the 20 mL of cells contained antibiotics but not xylose, we prepared the media with 2.2 mM (1.1x) xylose so the final concentration in the well would be 2 mM. The plate was then grown in a TECAN Spark plate reader with incubation and shaking with measurements taken every 10 minutes for up to 10 h. In all data shown, the measurements used were from the time point at which the well reach an OD of 1 on the Spark because this was at the end of exponential phase. For Fig. 3 the protocol was the same, but the media was prepared with LB, antibiotics, 2.2 mM (1.1x) xylose, and 1.1x the IPTG or ribose concentration for each graph. All replicates shown on all graphs are biological replicates carried out on different days. Relative fluorescence and strain ratio normalization were carried out using the equations in Table 1 below (example is for CFP, the same process was used for YFP).

**Confirming strain ratios**. To verify that the actual strain ratios matched the ratio at which the cultures were mixed, we developed a plating assay to count fluorescent colonies. In this system we sampled the wells from the 96-well plate after they reached the desired OD of 1 on the Spark. We diluted the contents of the wells by serial dilution to a final dilution of $10^{-6}$. We spread 75 mL onto a small (60 mm) LB agar plate with the appropriate antibiotics. This dilution concentration and plating volume were determined through multiple tests to be the best for even spread of colonies and for keeping colony counts between 50 and 150 which allowed the colonies to be spread out enough that they do not repress each other on the plate. The plates were grown 16–18 h at 37 C and then imaged using the same upright Nikon microscope with a ×0.5 objective as before. Using cyan and yellow fluorescence images of each plate from the microscope, we counted the number of yellow and cyan colonies on each plate to measure the strain ratios. This was done for 3–4 of the biological replicates in each graph of Figs. 2, 3, and 5b. The data from this is shown in Supplementary Fig. 5.

**Model for the majority wins pattern and inducer effect**. The logistic growth model[49]

$$\dot{N} = \lambda N \left(1 - \frac{N}{C}\right), \tag{1}$$

describes the population growth dynamics, where $N$ denotes the total number of cells in the liquid culture, $\lambda$ is the cell growth rate coefficient, and $C$ denotes the carrying capacity of the well. Logistic growth consists of an exponential phase followed by a saturation phase. Similar growth phases were observed in liquid culture: The saturation phase started about 5 h into the experiment when the population exhausted most nutrients, leading to cell death and the use of secondary carbon sources. As a result, cell growth rate slowed down, and cell count approached the carrying capacity $C$ (Supplementary Fig. 3 shows the optical density (OD) of cells in the wells across both phases). Population dynamics in the saturation phase is more complex than indicated by the logistic model. However, all reported experimental measurements were taken during the exponential phase, and we thus restrict our analysis to this phase.

The total population count can be written as $N = N_1 + N_2$, where $N_1$ and $N_2$ denote the number of yellow and cyan strain cells, respectively. By setting $r = N_1/N$ (the ratio of yellow strain cell count to total cell count), we have $N_1 = rN$ and $N_2 = (1 - r)N$. We assume that the ratio $r$ remains fixed at the value we set at the beginning of the experiment. We verified this assumption by counting the number of cells in the yellow and blue colonies taken from the liquid culture and plated on agar plates (Supplementary Fig. 5).

**Table 1 Normalization equations.**

| Normalization | Consortium | Equation |
| --- | --- | --- |
| Background subtraction (b.s.) | Both | =(CFP(AU))−(CFP(AU) of 0% cyan strain well) |
| Strain ratio normalization | Minority wins | =(CFP(b.s.))/(cyan strain ratio) |
| Relative fluorescence | Majority wins | =(CFP(b.s.))/(CFP(b.s.) of 100% cyan strain well) |
| Strain ratio normalization | Majority wins | =(relative CFP)/(cyan strain ratio) |

These equations were used to normalize the data from each well presented in Figs. 2b, c, e, f, 3, 5b and Supplementary Figs. 4 and 6. Since the minority wins strains are non-fluorescent in monoculture, they cannot be normalized to a relative fluorescence as the majority wins strains can relative to their own maximum fluorescence (in monoculture). Strain ratio normalization removes changes simply due to change in amount of cells expression the fluorescent protein. All data presented are at least background subtracted. Figures 2b, e and 3 follow the relative fluorescence normalization; Fig. 2b, e and and Supplementary Fig. 6 follow the strain ratio normalization for majority wins. Figure 5b follows the strain ratio normalization for minority wins. Plots in Supplementary Fig. 4 follow the relative fluorescence normalization for the majority wins and the strain ratio normalization for the minority wins. These normalization calculations can be carried out from the data in the Source Data File using Excel, MATLAB, or any similar program.

The differential equations that describe the dynamics of protein synthesis and degradation have the form

$$\dot{x}_1 = \alpha_0 \frac{1}{1 + \left(\frac{(1-r)Nx_2}{\theta_1} h(I)\right)^{n_1} + L_1 h(I)} - \beta x_1 - \frac{D_1 x_1}{Q + x_1}, \quad (2)$$

$$\dot{x}_2 = \alpha_0 \frac{1}{1 + \left(\frac{rNx_1}{\theta_2}\right)^{n_2} + L_2} - \beta x_2 - \frac{D_2 x_2}{Q + x_2}, \quad (3)$$

where

$$h(I) = \frac{1}{1 + \left(\frac{I}{K}\right)^\ell}, \quad (4)$$

$$\beta = \lambda \left(1 - \frac{N}{C}\right). \quad (5)$$

Equations (2) and (3) describe protein synthesis and degradation averaged over cells in the population. Here, $x_i$ denotes the concentration per cell of both synthase and fluorescent protein in strain $i$ ($i = 1, 2$). We assume that synthase concentration per cell equals fluorescent protein concentration per cell within each strain because the same promoters and QS molecules regulate the production of both synthase and fluorescent protein. While this assumption is wrong within individual cells due to random fluctuations, it is approximately correct when averaging over the population.

The terms $-\beta x_i$ ($i = 1, 2$) describe dilution, where $\beta$ is the instantaneous cell growth rate given in Eq. (5). The term $\frac{D_i x_i}{Q + x_i}$ captures enzymatic degradation of protein in strain $i$ ($i = 1, 2$). We obtain the functional form of enzymatic degradation using Michaelis–Menten dynamics[47].

The protein synthesis rate for strain 1 (the yellow strain) is a product of the maximal production rate, $\alpha_0$, and a regulatory function that takes into account the effects of inhibition by the QS molecule, inducer, and leakiness. We model protein synthesis rate in strain 2 in a similar fashion. Further details about the model are provided in the Supplementary Methods. In particular, Supplementary Table 2 lists the parameter values we used for simulations.

**Microfluidic experiments.** To perform the microfluidic experiments with either the majority or minority wins consortium, the yellow and cyan strains were each grown 12–18 h overnight in 5 mL of LB broth containing the appropriate antibiotics inoculated from a transformation plate. The next morning, the cultures were each diluted 1/1000 into 50 mL of fresh LB with antibiotics, still in mono-cultures. The two cultures were grown for about 3 h until they reached an OD of about 0.3–0.4. In the meantime, the microfluidic device was prepared by pre-warming to 37 C and flushing with 01% (v/v) Tween-20 to remove air from the device. Tween-20 is a surfactant and helps prevent cells from sticking to the channels of the device. Then a media syringe and two water syringes were prepared to connect to the proper ports of the device (Supplementary Fig. 8).

For the media, 20 mL of LB broth with 0.075% (v/v) Tween-20, 50 μg ML⁻¹ of kanamycin, 50 μg ML⁻¹ of spectinomycin, 33 μg ML⁻¹ of chloramphenicol, and 2 mM xylose was put into a 20 mL syringe without the plunger and attached to the media port of the device. For the two waste ports, two 10 mL syringes without the plungers were filled with sterile water and connected to each waste port. Once the cells reached an OD of about 0.3–0.4, 15 mL of each culture was spun down at 2000 × g for 5 min. The media was removed from each and both pellets were resuspended together in a final volume of 10 mL of pre-warmed (to 37 C) LB with antibiotics. This resuspension allows for a higher concentration of cells while still being in exponential growth. The resuspended cells were put into a 10 mL syringe without the plunger and then attached to the cells port of the device.

The heights of the syringes without the plungers (due to gravity) determines the flow of liquid from the syringes through the device. The two water waste syringes are always the lowest and the media syringe is always the highest. To load cells into the trap, we raise the cell syringe almost to the height of the media allowing for cells to reach the traps but not to contaminate the media port. We leave the syringes at this height for 30 min to an hour to fill as many cells into the traps as possible. This

increases our chances of capturing both strains in each trap. Then we lower the cell syringe so that fresh media reaches the cell traps. We allow the cells to grow and acclimate to the device for up to 2 h then move to the higher ×100 oil objective on an inverted Nikon fluorescence microscope. We image every trap under phase contrast, yellow fluorescence, and cyan fluorescence every 6 minutes for up to 24 h. To obtain the images, we export tiff images for each channel of each trap over time. Construction of the microfluidic devices and overall set up procedure were adapted from Ferry et al.[50].

In ImageJ we compiled the phase contrast, yellow, and cyan channels to create the images shown in Fig. 6. Data analysis to provide the graphs from Fig. 6 and Supplementary Fig. 9 was performed with custom MATLAB code. This code determined the strain identity of cells based on the intensity of yellow and cyan fluorescence at each pixel. Thresholds are dynamic over time: for example, when cyan is ON, thresholds are set so that any pixel above a set cyan threshold belongs to the cyan strain and any pixel below that threshold belongs to the yellow strain. When both strains are dim, a threshold ratio of yellow to cyan fluorescence is set to label the pixels. These pixels are labeled in a mask output by MATLAB and compared to the images to confirm they are as close to the true strain identities as possible. Total pixels identified as the yellow or cyan strain are quantified and used to calculate the estimated strain ratios presented in the graphs. The yellow and cyan strain masks are applied to the raw yellow and cyan fluorescence values and averaged to get the average fluorescence values presented in the same graphs.

**Reporting summary.** Further information on research design is available in the Nature Research Reporting Summary linked to this article.

## Data availability

All data generated or analyzed during this study are included in the Source Data file. Any additional relevant data are available from the authors upon reasonable request. Plasmids are available through Addgene (ID 141123-141130).

## Code availability

The code for analyzing microscopy data uses MATLAB and is available on Github: https://github.com/razanalnahhas/-Majority-sensing-in-synthetic-microbial-consortia.

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

## Acknowledgements

This work has been partially supported by NIH grant RO1 GM117138 (W.O., K.J., M.R.B.); NSF grants DMS 1816315 (W.O.), DMS 1662305 (K.J.), MCB 1936770 (K.J.), MCB 1936774 (M.R.B.), DMS 1662290 (M.R.B.), and GRFP 1842494 (R.N.A.); and the Welch Foundation grant C-1729 (M.R.B.).

## Author contributions

R.N.A. designed consortia, performed experiments, and analyzed data. M.S., W.O., and K.J. developed the models. Y.C. and R.N.A. built plasmids. A.A.F. helped with experiments. M.R.B. oversaw the project. All authors contributed to discussion and development of the project and helped write the manuscript.

## Competing Interests

The authors declare no competing interests.
