## [Peer Review File · Nature Communications]

Review of the paper: Majority Sensing in synthetic microbial consortia

Summary

This paper focuses on the development of novel tools for gene expression in microbial co-cultures. The authors built a majority-win system and a minority-win system, both of which can vary their gene expression profiles based on the proportion of the two co-cultured strains. The paper shows that the expression of a fluorescent protein can be either turned ON or OFF by a communication molecule produced by a different strain at a given ratio in the consortium. Additionally, the behaviours of the consortia can be finely controlled by externally adding inducers. The behaviours of the various consortia have been demonstrated across three growth conditions: on solid agar, in small batch culture in 96-well plates and in a continuous culture in a microfluidic chamber. The paper also presents a simple model that is able to predict the fluorescence levels of both strains when the consortium is seeded with different initial strain ratios or supplemented with different inducers. Although the standard deviations of the systems appear to be quite large, the means taken across all biological replicates agree with the modelling results.

General comments

Each figure would benefit from displaying the number of biological replicates used for each experiment. Whenever more than 3 biological replicates were used, it would be interesting to know why and how this reflects on the robustness and reproducibility of the tools developed in the paper. Providing more details on how the system was built would be useful for the reader to comprehend the properties of the systems. Currently no information is given on how the constructs were assembled, and how the different parts were selected for the final design. Sequences of the genetic parts used in the circuits are not provided. It is imperative for reproducibility that all sequences are provided and properly annotated.

To understand how usable the system proposed actually is for consortia-based expression of different genes or metabolic pathways, it is important to investigate whether the co-repressive consortia presented in the paper would work if the genes of interest differ more than just by a few point mutations. I would thus suggest that the author test the system on one or two genes of interests or pathways that are substantially different beyond those currently considered.

Specific Comments

The following points and questions are brought to the attention of the reviewer and should be discussed as part of the main paper and/or supplementary material before the paper can be further considered for publication.

1. **Page 3 - Results:**

- How were C4-HSL and C14-HSL selected as suitable candidate for the system? Data from Kyllis et al. (2018) [1] showed that C4-HSL and C14-HSL have different input-output static response curves, which could be a problem for tuning the system. Was this a problem for building up the systems that this paper is presenting? If so, was the problem addressed by tuning the regulatory elements of the quorum sensing components of the circuit? Please provide an explanation for this to help justify the design and tuning of the system.

- How do you justify the need for using strong degradation tags and a lactonase degrading C4-HSL and C14-HSL in the system? Was the choice informed by the modelling or previous experimental work? Please provide an explanation for this to help justify the design and tuning of the system.

[1] Kylilis, N., Tuza, Z.A., Stan, G.B. and Polizzi, K.M., 2018. Tools for engineering coordinated system behaviour in synthetic microbial consortia. *Nature communications*, 9(1), pp.1-9.

2. **Page 5 - Figure 2:** How was the normalisation for the strain fraction applied? Perhaps displaying the formula could help the reader with the understanding of the (c) and (f) plots more easily.
3. **Page 6 (lines 124-125) - Majority Wins Consortium:** Did you expect the fluorescent protein expression to be different for the co-repressive consortium and the control consortium? Did the authors compare the strength of the pLac and the pIQ promoters? Again, please provide an explanation for this to help justify the design and tuning of the system.
4. **Page 7 - Figure 3:**
 - Please include the number of biological replicates in the figure caption (n=...)?
 - Would it be more representative to display the relative fluorescence/strain fraction instead of the relative fluorescence as you did in Figure 2 (c) and (f)? This would show that again the increase or decrease of fluorescence is not due to a change in cell numbers.
 - It would be useful and informative to add the reference to Supplementary Figure S1 to show how the model developed in the paper can predict the cross-over point by tuning the level of inducer.
5. **Page 9 - Figure 4:**
 - Please the number of biological replicates in the figure caption (n=...)?
 - How do you explain that the cyan relative fluorescence always collapses to 0 while the model predicts that it should settle to a positive value when the yellow strain fraction is small? Please explain.
6. **Page 11 - Figure 5 (b):** How does the paper explain the large standard deviation in the biological replicates in (b)? Please include the number of biological replicates in the figure caption (n=...)?
7. **Page 12:** Why does the ratio of cells change through time if you have shown in Figure S3 that the different strains have a similar growth rates? Have you been able to observe ratio fluctuations during the time course of the plate-reader experiments in 96-well plates? Please explain.
8. **Page 11 - Co-repressive consortium plasmids:**
 - Why was the sdiA knock-out required for building the co-repressive consortium? Please provide an explanation for this to help justify the design and tuning of the system.
 - Please include the plasmid maps of the constructs in the supplementary alongside with the sequences of the genetic parts used in this paper.
 - What motivated the choice of using bicistronic RBSs? Was it to increase enzyme production? Please provide an explanation for this to help justify the design and tuning of the system.
9. **Page 17 - Confirming strain ratios:** The use of a flow cytometer could help quantifying the ratio of the two strains in the consortium in a more rapid and high-throughput manner. Does the method described in the paper for quantifying strain ratios present advantages compared to using a flow cytometer, assuming that as Figure S3 shows, the cells do not suffer from toxicity effects caused by the genes being expressed?
10. **Supp. Page 9 - Figure S5:**
 - How was the normalized fluorescence calculated in both graphs? Please explain this more thoroughly.
 - The fluorescence level of the yellow strain (when induced) seems to be around 8 times higher than that of the cyan strain. Why are the normalised fluorescence levels so different in the yellow and cyan strains? Please explain.

Reviewer #2:

Remarks to the Author:

In this paper, Razan et al. developed a two-strain co-repressive gene circuit that can sense the majority in microbial consortia. The system is composed of two strains, each of which produces an intercellular signaling molecule to tune down the production of its orthogonal molecules. The circuits are carried by two plasmids. Each plasmid contains three elements: an acyl-homoserine lactone synthase (either RhlI or CinI), a transcriptional repressor (LacI or RbsR-L), and a fluorescent reporter (sfCFP or sfYFP). The transcriptional repressor RbsR-L responds to the quorum sensing (QS) molecules C14-HSL, while LacI responds to C4-HSL. Therefore, when the two strains are cultured together, the more numerous strain will produce more QS molecules overall and shut down the QS production in the other strain.

First, Razan et al. tested them in monoculture, where addition of the opposite QS molecule turned the fluorescence of each strain 'OFF'. Then, they mixed two strains in 10% increments on agar plate or in liquid media, and quantify the fluorescence of the mixture. The results suggested the dominance of majority in the fluorescence pattern. Next, they showed that the crossover point at which one strain wins can be tuned by external inducer IPTG or ribose. They developed a mathematical model to describe the dynamic behavior of this system. They also changed the promoters of the co-repressive consortium to engineer a 'minority win' consortium. In this consortium, the fluorescence was directly induced by the QS molecules produced by the opposite strain. At last, using microfluidic devices, they showed that both systems responded to the fluctuating strain fractions.

General comments

Quorum sensing has been well-established as a mechanism by which bacteria sense and respond to changes in their population sizes. This study exploits this property and uses density sensing as an important parameter to amplify the dynamics of gene circuits. For instance, in the majority sensing system, the relative fluorescence (per unit biomass) is critically dependent on the strength of mutual inhibition, which in turn is sensitive to the densities of the two strains.

Overall, the results (both modeling and experiments) are comprehensive and convincing in terms of demonstrating the concept of majority sensing (or minority sensing). I believe the paper can be strengthened by better articulating when and why the majority sensing (or minority sensing) can be important or useful. For instance, as noted by authors, at the consortium level, the total fluorescence of the consortium depends on the ratio of two strains, with or without the co-repressive switch. The key difference is that the single-gene expression in the co-repressive switch responds to the change in the ratio; single-cell gene expression in the consortium without the co-repressive switch is independent of the strain ratio. As such, it is important to explain why this difference is valuable in a broader context.

Other points

(1) As noted above, a defining feature of the majority (or minority) sensing that the authors emphasize is the average response from single cells (e.g., the contrast between Figure 2c and Figure 2f). Thus, it seems more consistent by presenting the data in Figures 3 & 4 in the same manner as Figure 2c and Figure 2f.

(2) As it stands, the modeling component of the work seems to primarily provide a consistency check for experimental data interpretation. I wonder if the additional insights can be gained about certain quantitative properties of the ratio sensing.

(3) For the microfluidics experiments, it was tricky to determine the identity of the cells when the fluorescence is off. Here, the author used the following strategy: 'when cyan is ON, thresholds are set so that any pixel above a set cyan threshold belongs to the cyan strain and any pixel below that threshold belongs to the yellow strain. When both strains are dim, a threshold ratio of yellow to cyan fluorescence is set to label the pixels', which can induce a lot of errors. Will it be better to label one the strains with constitutively expressed fluorescence like mCherry, so that the two strains can be better distinguished from each other.

(3) I am curious whether this idea of design can be generalized to more complex communities that are composed of multiple members instead of two.

We thank the reviewers for their thorough reading of our manuscript and insightful comments. We have subsequently rewritten the manuscript to address the issues raised by the reviewers. A point-by-point response (in blue) to each comment is given below.

Reviewer 1

General comments

Each figure would benefit from displaying the number of biological replicates used for each experiment. Whenever more than 3 biological replicates were used, it would be interesting to know why and how this reflects on the robustness and reproducibility of the tools developed in the paper.

Thank you for your comments. The number of replicates has been added to the captions of each figure.

Providing more details on how the system was built would be useful for the reader to comprehend the properties of the systems. Currently no information is given on how the constructs were assembled, and how the different parts were selected for the final design. Sequences of the genetic parts used in the circuits are not provided. It is imperative for reproducibility that all sequences are provided and properly annotated.

Each of the components within the plasmids used in this system (and the strain) were described in detail in our previous paper (Chen et al., Science 349, 986 (2015)). For completeness, we have included the details of the plasmids again in the supplementary material.

To understand how usable the system proposed actually is for consortia-based expression of different genes or metabolic pathways, it is important to investigate whether the co-repressive consortia presented in the paper would work if the genes of interest differ more than just by a few point mutations. I would thus suggest that the author test the system on one or two genes of interests or pathways that are substantially different beyond those currently considered.

We believe that the current set of experiments is convincing. There is extensive literature in which fluorescent reporters were used as easily quantifiable proxies for downstream targets of synthetic gene circuits. To be sure, some idiosyncratic retroactivity could occur depending upon the choice of downstream targets. However, retroactivity of this sort has been extensively studied and is beyond the scope of this project.

Specific Comments

The following points and questions are brought to the attention of the reviewer and should be discussed as part of the main paper and/or supplementary material before the paper can be further considered for publication.

1. Page 3 - Results:

– How were C4-HSL and C14-HSL selected as suitable candidate for the system? Data from Kyllis et al. (2018) [1] showed that C4-HSL and C14-HSL have different input-output static response curves, which could be a problem for tuning the system. Was this a problem for building up the systems that this paper is presenting? If so, was the problem addressed by tuning the regulatory elements of the quorum sensing components of the circuit? Please provide an explanation for this to help justify the design and tuning of the system.

[1] Kylilis, N., Tuza, Z.A., Stan, G.B. and Polizzi, K.M., 2018. Tools for engineering coordinated system behaviour in synthetic microbial consortia. *Nature communications*, 9(1), pp.1-9.

C4-HSL and C14-HSL were chosen primarily because their respective signaling systems exhibit little to no crosstalk in our *sdiA* knock-out strain. Further, as evidence by our previous work (Chen et al. *Science* 349, 986 (2015) and Kim et al. *Nature Chem Biol* 15, 1102 (2019)), the two systems work very well together. The difference in the static response curves of sensor systems to the two HSLs is somewhat irrelevant, as the relative concentrations needed for activation of downstream components of one system has no bearing on how the other operates. What matters more is how well a synthetic system can produce and respond to any given HSL. The strain we used contains chromosomally integrated copies of *cinI* and *rhII*, which encode the enzymes that make their respective HSLs. We have found (again evidenced by Chen et al. *Science* 349, 986 (2015) and Kim et al. *Nature Chem Biol* 15, 1102 (2019)) that these enzymes produce enough HSLs to elicit downstream signals. Some tuning of the target promoters was needed to optimize input-output relations (described in Chen et al. *Nature Comm* 9, 64 (2018)). Note, however, that this tuning was done to increase the output sensitivities of the two QS systems in isolation to one another, and not as a result of an imbalance in the necessary concentrations of HSLs between the two.

– How do you justify the need for using strong degradation tags and a lactonase degrading C4-HSL and C14-HSL in the system? Was the choice informed by the modelling or previous experimental work? Please provide an explanation for this to help justify the design and tuning of the system.

The degradation tags and the lactonase were originally included on these plasmids to decrease the characteristic timescales of circuits built with them. While this is important for circuits like dual-feedback oscillators, it is actually not important for the current circuit. However, the removal of the lactonase and deg tags would disrupt the steady state levels of the key components of the system and hence potentially alter input-output relationships. Therefore, because the input – output relations of the two QS systems are tuned well already, we decided to keep the degradation tags and lactonase. We have added this explanation to the main text methods (lines 271-275).

2. Page 5 - Figure 2: How was the normalisation for the strain fraction applied? Perhaps displaying the formula could help the reader with the understanding of the (c) and (f) plots more easily.

You are absolutely right. We have added the following formulas to the methods. The first step removes background fluorescence from the media and cell autofluorescence. The second step (for the majority wins) sets the fluorescence relative to each maximum in order to directly compare *cfp* and *yfp*. The last step for both normalizes to strain ratio in order to quantify the relative single cell expression level.

background subtracted fluorescence (all samples)
= (CFP intensity of each well) – (CFP intensity of yellow strain only well)

relative fluorescence (majority wins only)
=
$$\frac{\text{background subtracted CFP of each well}}{\text{background subtracted CFP of cyan strain only well}}$$

strain ratio normalization (majority wins): $\frac{\text{relative CFP of each well}}{\text{cyan strain ratio}}$

strain ratio normalization (minority wins): $\frac{\text{background subtracted CFP of each well}}{\text{cyan strain ratio}}$

3. Page 6 (lines 124-125) - Majority Wins Consortium: Did you expect the fluorescent protein expression to be different for the co-repressive consortium and the control consortium? Did the authors compare the strength of the pLac and the pIQ promoters? Again, please provide an explanation for this to help justify the design and tuning of the system.

We expected the expression levels to be different, as the genes encoding the fluorescent proteins are different. It is exceedingly hard to match transcription rates from promoters that have different regulatory inputs. We know that the control consortium fluorescence is dimmer than the co-repressive consortium fluorescence, since the pIQ promoter is weaker than the pLac promoter when induced. The difference in intensity of the promoters of the two consortia is not an issue as we normalized both to their own maximum (see above) then compared the relative patterns.

4. Page 7 - Figure 3:

– Please include the number of biological replicates in the figure caption (n=...)?

They have been added, thank you for the reminder.

– Would it be more representative to display the relative fluorescence/strain fraction instead of the relative fluorescence as you did in Figure 2 (c) and (f)? This would show that again the increase or decrease of fluorescence is not due to a change in cell numbers.

We debated this as well. We have now added the normalized version to the supplement (Fig. S6), so both are available to readers.

– It would be useful and informative to add the reference to Supplementary Figure S1 to show how the model developed in the paper can predict the cross-over point by tuning the level of inducer.

We have added this reference, thank you for the recommendation.

5. Page 9 - Figure 4:

– Please the number of biological replicates in the figure caption (n=...)?

They have been added, thank you for the reminder.

– How do you explain that the cyan relative fluorescence always collapses to 0 while the model predicts that it should settle to a positive value when the yellow strain fraction is small? Please explain.

Our model was meant to capture the essential mechanisms that drive the observed dynamics. We opted for a simple, easily understandable model, rather than one that would accurately reproduce nuances in the response. The reason for this was that a complex model would be

hard to validate, and while more flexible, would also be harder to tune and understand. Thus, while tractable, our model only reproduces experimental results qualitatively.

We could tune and extend the model to better match experimental data: The sensitivity of the Hill functions used in the model could be changed, or the cascade of intermediate reactions that model the interaction between strains could be modeled more accurately. A preliminary analysis suggests that such increased flexibility could result in a model that better matches the experimental result.

However, given the challenge in obtaining experimental replicates, and the trial-trial variability, we have opted not to increase model complexity. For instance, it is also possible that the collapse is not immediate in the experiment, but we don't have the resolution to show this. Therefore, given the data we opted to present the simplest model that explains the circuit's behavior.

6. Page 11 - Figure 5 (b): How does the paper explain the large standard deviation in the biological replicates in (b)? Please include the number of biological replicates in the figure caption (n=...)?

The large standard deviation in Fig. 5(b) could be due to several things. For instance, it could be that the pCin promoter used in the minority wins circuit is simply noisier than its pLac counterpart in the majority wins circuit. There could also be some retroactive effect that occurs due to changing the number of LacI and CinR operator sites in the system. We believe that tracking down the source of variability is beyond the scope of this paper, however.

The number of biological replicates has been added, thank you for the reminder.

7. Page 12: Why does the ratio of cells change through time if you have shown in Figure S3 that the different strains have a similar growth rates? Have you been able to observe ratio fluctuations during the time course of the plate-reader experiments in 96-well plates? Please explain.

The fluctuations in the strain ratio in the microfluidic device occur due to instabilities in the growth pattern within the chamber and not from differences in the growth rates. We have explored this phenomenon in a previous paper that we cite (Alnahhas et al. Spatiotemporal dynamics of synthetic microbial consortia in microfluidic devices. ACS Synthetic Biology, 8(9):2051-2058, 2019). We have made this clearer in the text starting at line #207:

“To do so, we chose the ‘Hallway’ microfluidic device that has small cell trapping chambers off to the side of the media flow channel, as shown in Supplementary Figure 8. This device has previously been demonstrated to result in fluctuations of strain ratios over time due to the small population size within the cell trapping region [38].”

Note that such drastic fluctuations in the strain ratio are not observed in plate reader experiments. To confirm this, we sampled the plate reader experiments at beginning and end and counted yellow and cyan colonies (Supplementary Figure 5) and saw no significant changes over time.

8. Page 11 - Co-repressive consortium plasmids:

– Why was the sdiA knock-out required for building the co-repressive consortium? Please provide an explanation for this to help justify the design and tuning of the system.

The *sdiA* gene encodes a transcription factor that activates the RhIR-responsive promoter using other HSLs (Lindsay et al. "Effect of *sdiA* on biosensors of N-acylhomoserine lactones." J Bacteriol 187, 5054-5058 (2005)). Our previous work has shown that knocking out *sdiA* reduces cross talk when using the CinI/R and RhII/R systems together in *E. coli*. We've included this in the methods line 274 (Chen et al. Science, 349(6251):986-989, 2015).

– Please include the plasmid maps of the constructs in the supplementary alongside with the sequences of the genetic parts used in this paper.

These have been added to the supplement and referred to in the paper (line 270). Thank you for the suggestion.

– What motivated the choice of using bicistronic RBSs? Was it to increase enzyme production? Please provide an explanation for this to help justify the design and tuning of the system.

The BCD element allows for the strength of the RBS not to be influenced by the downstream sequence, therefore keeping relative strengths between the two strains the same (Mutalik et al. "Precise and reliable gene expression via standard transcription and translation initiation elements." Nat. Methods, 10(4):354-360, 2013.). Again, we have found from our previous work that when used in these CinI/R-RhII/R consortial systems the resulting regulatory levels work very well.

9. Page 17 - Confirming strain ratios: The use of a flow cytometer could help quantifying the ratio of the two strains in the consortium in a more rapid and high-throughput manner. Does the method described in the paper for quantifying strain ratios present advantages compared to using a flow cytometer, assuming that as Figure S3 shows, the cells do not suffer from toxicity effects caused by the genes being expressed?

The problem with using a flow cytometer is that non-fluorescent cells cannot be identified. By plating dilutions, the colonies are separated sufficiently, so they can't repress each other, and are therefore fluorescent and identifiable (details in lines 345-360 of main text).

10. Supp. Page 9 - Figure S5:

– How was the normalized fluorescence calculated in both graphs? Please explain this more thoroughly.

The normalization here was actually a background subtraction. This has been clarified in the caption.

– The fluorescence level of the yellow strain (when induced) seems to be around 8 times higher than that of the cyan strain. Why are the normalised fluorescence levels so different in the yellow and cyan strains? Please explain.

Normalization was not the best label, since it's just a background subtraction and we've corrected the wording. Thank you for noticing! The difference in intensity is due to the different quantum efficiency of the fluorophores.

Reviewer #2 (Remarks to the Author):

General comments

Overall, the results (both modeling and experiments) are comprehensive and convincing in terms of demonstrating the concept of majority sensing (or minority sensing). I believe the paper can be strengthened by better articulating when and why the majority sensing (or minority sensing) can be important or useful. For instance, as noted by authors, at the consortium level, the total fluorescence of the consortium depends on the ratio of two strains, with or without the co-repressive switch. The key difference is that the single-gene expression in the co-repressive switch responds to the change in the ratio; single-cell gene expression in the consortium without the co-repressive switch is independent of the strain ratio. As such, it is important to explain why this difference is valuable in a broader context.

Thank you for your comments. We have expanded the explanation of the significance and provided example applications in the Discussion section (lines 249-257):

“Possible applications of the majority wins phenotype include population control by encoding a toxin gene within the circuit. Then if a strain begins to overgrow and become a majority, the toxin could be expressed and reduce the growth rate of this strain until a balance is reached again. A possible use of the minority wins phenotype would be in bioprocessing. If a biochemical pathway is split between strains within a consortium and one strain begins to overgrow there would be a buildup and bottleneck in the pathway. If the minority wins genetic circuit is adapted to this pathway, then the system would down-regulate gene expression in the overgrowing strain to prevent a buildup while also up-regulating gene expression in the minority strains preventing a backup at that step of the process.”

Other points

(1) As noted above, a defining feature of the majority (or minority) sensing that the authors emphasize is the average response from single cells (e.g., the contrast between Figure 2c and Figure 2f). Thus, it seems more consistent by presenting the data in Figures 3 & 4 in the same manner as Figure 2c and Figure 2f.

Figure 2f shows the control consortium, which does not respond to inducers. Figures 3&4 are response to inducers, so there would not be anything to compare to with the control consortium. If you were referencing the strain ratio normalization, we have added that in the supplement (Supplementary Figure 6).

(2) As it stands, the modeling component of the work seems to primarily provide a consistency check for experimental data interpretation. I wonder if the additional insights can be gained about certain quantitative properties of the ratio sensing.

We agree that the main point of the model was to confirm the basic mechanisms of interaction both in circuits in which the majority and the minority win. The fact that the model captures the qualitative behavior of the response without fine-tuning suggests that it does describe the essence of strain interactions. However, we found that one main prediction of the model that was not discussed enough is that the majority/minority sensing is roughly independent of the size of the colony (provided there are enough cells to create a large enough signal and the

system is well-mixed). To better discuss this feature, we have added new experiments that provide further evidence for size-independence (Supplementary Figure 4).

We have also added the following text to the discussion of the model: “The model did not explicitly include the number of cells as a parameter, predicting that behavior should be independent of colony size. This prediction was confirmed in separate experiments (Supplementary Figure 4).”

(3) For the microfluidics experiments, it was tricky to determine the identity of the cells when the fluorescence is off. Here, the author used the following strategy: ‘when cyan is ON, thresholds are set so that any pixel above a set cyan threshold belongs to the cyan strain and any pixel below that threshold belongs to the yellow strain. When both strains are dim, a threshold ratio of yellow to cyan fluorescence is set to label the pixels’, which can induce a lot of errors. Will it be better to label one the strains with constitutively expressed fluorescence like mCherry, so that the two strains can be better distinguished from each other.

This is a good suggestion thank you, and we considered it as well. However unfortunately there is not a fourth non-overlapping fluorescence we could use in order to label both strains. Further, we were concerned about creating an imbalance in the system by labeling only one strain. While not perfect, we believe our estimates of the strain ratios within the microfluidic devices are good enough to qualitatively understand the dynamics of the systems.

(3) I am curious whether this idea of design can be generalized to more complex communities that are composed of multiple members instead of two.

This is an interesting idea we would consider for a follow up paper – and something we very much want to do! However, given the current situation we cannot carry out any further experiments at the moment, and believe such experiments to be outside the scope of the current manuscript.

Reviewers' Comments:

Reviewer #1:

Remarks to the Author:

The authors have addressed my major concerns and made additions to the main text and supplementary file accordingly. Thank you!

The only point that I am still a little concerned by is the large error bars (point 6, Page 11 - Figure 5 (b))

Reviewer #2:

Remarks to the Author:

The authors have satisfactorily addressed my raised points.

Response to Reviewer Requests (in purple):

REVIEWERS' COMMENTS:

Reviewer #1 (Remarks to the Author):

The authors have addressed my major concerns and made additions to the main text and supplementary file accordingly. Thank you!

Thank you!

The only point that I am still a little concerned by is the large error bars (point 6, Page 11 - Figure 5 (b))

The fluorescence of the minority wins strains is low when measured in bulk fluorescence since only a few cells (the minority) are fluorescent. Due to this low level, noise makes a more significant effect. We believe this to be the source of this large distribution and have explained that in the text as well. Thank you for suggesting this point of clarification.

Reviewer #2 (Remarks to the Author):

The authors have satisfactorily addressed my raised points.

Thank you!